

# Aetiology of livestock fetal mortality in Mazandaran province, Iran

Afsaneh Amouei[1,2,3], Mehdi Sharif[1,3], Shahabeddin Sarvi[1,3], Ramin Bagheri Nejad[4], Sargis A. Aghayan[5], Mohammad Bagher Hashemi-Soteh[6], Azadeh Mizani[1], Seyed Abdollah Hosseini[1,2,3], Sara Gholami[3], Alireza Sadeghi[7], Mohammad Sarafrazi[8] and Ahmad Daryani[1,3]

[1] Toxoplasmosis Research Center, Mazandaran University of Medical Sciences, Sari, Mazandaran, Iran
[2] Student Research Committee, Mazandaran University of Medical Sciences, Sari, Mazandaran, Iran
[3] Parasitology Department, Faculty of Medicine, Mazandaran University of Medical Sciences, Sari, Mazandaran, Iran
[4] Brucellosis Department, Razi Vaccine and Serum Research Institute, Agricultural Research, Education and Extension Organization, Karaj, Alborz, Iran
[5] Laboratory of Zoology, Research Institute of Biology, Yerevan State University, Yerevan, Yerevan, Armenia
[6] Department of Clinical Biochemistry and Genetics, Faculty of Medicine, Mazandaran University of Medical Sciences, Sari, Mazandaran, Iran
[7] Mazandaran Central Laboratory of Veterinary Organization, Medical Sciences, Sari, Mazandaran, Iran
[8] Mazandaran Provincial Veterinary Department, Medical Sciences, Sari, Mazandaran, Iran

Corresponding author
Ahmad Daryani, daryanii@yahoo.com

## ABSTRACT

In the farming industry, the productivity of livestock herds depends on the fertility efficiency of animals. The accurate diagnosis of a broad range of aetiological agents causing fetal death is often difficult. Our aim was to assess the prevalence rates of *Toxoplasma gondii*, *Neospora caninum*, and *Brucella* spp. infections in ruminant abortion using bacteriological culture and molecular techniques in Mazandaran Province, northern Iran. Samples were collected from 70 aborted sheep, goat, and cattle fetuses between September 2014 and December 2015. Necropsy was performed on all the received samples, and brain tissue and abomasal content were obtained from the aborted fetuses. Protozoan infections were detected by specific polymerase chain reaction (PCR) and bacterial agents using bacteriological examinations and PCR assay. Infectious pathogens were detected in 22 out of 70 (31.4%) examined fetuses. Moreover, *T. gondii*, *N. caninum*, and *B. melitensis* were verified in 13 (18.6%), four (5.7%), and two (2.85%) samples, respectively. Our results showed that infection with the mentioned pathogenic agents may lead to fetal mortality, which can be a major cause of economic loss. The listed pathogens could be considered important etiological agents of fetal loss in Mazandaran Province, for which appropriate control measures such as vaccination and biosecurity can be implemented to prevent infection and reduce reproductive loss in livestock farms.

## INTRODUCTION

The productivity of livestock herds depends substantially on their reproductive efficiency. High fetal mortality rate is a major cause of economic loss in the farming industry, and a

broad range of protozoa, bacteria, and viruses is reported from ruminant farms. Therefore, the definitive diagnosis of abortifacient infectious agents is often difficult and should be established in specialized laboratories. Several causative pathogenic agents are considered as the potential sources of zoonotic infections that are of veterinary and public health significance (*Moeller Jr, 2001*).

*Toxoplasma gondii* and *Neospora caninum* are well-known protozoa causing congenital infections related to abortion, neonatal mortality, and necrotic lesions in the central nervous system (*Müller et al., 1996*). These parasites belong to the Apicomplexa phylum, which are morphologically analogous but have some structural, molecular, and antigenic differences. Their life cycles are also the same with diverse definitive hosts, such as felids and canids in *T. gondii* and *N. caninum*, respectively. They have similar intermediate hosts including a wide range of warm-blooded animals. *T. gondii* is also an important pathogen of humans that can be transmitted through handling or consumption of raw or uncooked meat and milk (*Hutchison, 1965*; *Tenter, Heckeroth & Weiss, 2000*). Neosporosis, caused by *N. caninum*, was first diagnosed in 1990 and was found a leading cause of abortion in cattle (*Wouda et al., 1997*). Moreover, *N. caninum* infection, which is associated with epizootic abortion, occurs less frequently in small ruminants (*Moreno et al., 2012*). Although antibodies against *N. caninum* were identified in humans, this parasite has not been isolated from human tissues (*Ibrahim et al., 2009*; *Lobato et al., 2006*). Fetal brain tissue injuries are similar to those in *T. gondii* and *N. caninum* infections; these lesions may be sparse and not easily found. Nevertheless, the diagnosis of these two coccidian parasites was improved by the development of polymerase chain reaction (PCR) assays (*Bretagne et al., 1993*; *Yamage, Flechtner & Gottstein, 1996*).

*Brucella* is the most important abortifacient bacterial agent with great economic importance in livestock in many areas of the world. Brucellosis, infection with the members of the genus *Brucella*, is still a widespread zoonotic disease known as a significant threat to human and animal health worldwide (*Whatmore, 2009*). Abortion is the most important clinical sign of the disease in infected female animals, which usually occurs in late pregnancy (*Radostits et al., 2006*). However, symptoms of the disease are mostly not pathognomonic; thus, its accurate and direct diagnosis depends on bacteriological tests (*Blasco, 1992*).

Despite the reports showing implication of *Brucella* spp. in small ruminants' abortion (*Behroozikhah et al., 2012*), limited epidemiological information is available as to the current frequency of abortion caused by *Brucella* in animal population in Iran.

The main objective of this study was to provide data about the occurrence of *T. gondii*, *N. caninum*, and *Brucella* spp. infection in cases of ruminant abortion (i.e., sheep, goats, and cattle) in Mazandaran Province, northern Iran.

## MATERIALS AND METHODS

### Study area

This study was carried out in Mazandaran Province, near the Caspian Sea, in the northern region of Iran, where the geographic and natural climatic conditions (temperature and humidity with an annual rainfall of 500 mm and an average temperature of 17 °C) are

suitable for livestock production. Data regarding the history of each farm, including epidemiological area code, abortions, and results of serological surveys for brucellosis using tests such as Rose Bengal Test (RBT), serum agglutination test (SAT), and 2-Mercaptoethanol (2ME) (*Alton et al., 1988*; *World Organization for Animal Health , OIE*) were obtained by interviewing the herders and by the examination of computerized herd records at the Central Laboratory of the Department of Veterinary Medicine in Mazandaran Province.

## Sample collection

Between September 2014 and December 2015, 70 aborted and dead fetuses (sheep, goat, and cattle) were provided by the Department of Veterinary Medicine in Mazandaran Province. All the investigations reported here were approved by the Ethics Committee of Mazandaran University of Medical Sciences (No. 1055). Data related to each animal were recorded using three independent variables (i.e., region, age, and animal species).

Data were obtained from at least one animal on each farm and from at least two different types of samples from the same animal. Necropsy was conducted on all the aborted fetuses. Samples of brain tissue and abomasal content were obtained from all the cases of abortion, stillbirth, and neonatal death that took place on different farms located in several areas (i.e., east, center, and west), which were all in Mazandaran Province. Different parts of the brain (e.g., cortex, midbrain, medulla, and cerebellum) were preserved in 70% ethanol until use for the PCR detection of *T. gondii* and *N. caninum*.

## Bacteriological examinations

For the isolation of *Brucella* spp., samples of fetal abdominal content from the aborted fetuses were cultured using standard, previously described procedures (*Alton et al., 1988*). Briefly, fresh specimens were cultivated onto *Brucella* medium base (OXOID, CM169B) containing *Brucella* Selective Supplement (OXOID, SR0383A) and 5% horse serum. The plates were incubated at 37 °C in an atmosphere of 10% $CO_2$. *Brucella* was recognized by colony morphology, growth, culture, staining, and biochemical characteristics such as oxidase and urease. Species and biovar of the isolated *Brucella* strains were determined by the standard methods including $CO_2$ requirement, $H_2S$ production, agglutination with mono-specific antisera, susceptibility to fuchsin and thionin dyes, and lysis by Tb phage (*Alton et al., 1988*).

Samples for the isolation of other microorganisms were also inoculated onto Blood Agar containing 7% defibrinated sheep blood, MacConkey Agar, *Eosin Methylene Blue (EMB) Agar, Salmonella/Shigella* Agar, Buffered Peptone Water (BPW) as pre-enrichment media, Tetrathionate (TT) and Selenite Broths as enrichment media.

## DNA extraction

For the detection of *Toxoplasma* and *Neospora*, DynaBio DNA Extraction Kit (Takapouzist co, Iran) was used to extract DNA from 20-mg brain tissue samples of all the aborted fetuses in accordance with the manufacturer's protocol.

For the *Brucella* isolates, PCR was performed on heat-killed cell suspensions. For this purpose, a loopful of bacterial cells was suspended in 200 µl of phosphate-buffered saline

(PBS) and boiled for 10 min. The suspension was then centrifuged at 12,000 rpm for 5 min and the supernatant was used as template DNA.

The concentration of DNA was estimated by spectrophotometric analysis at A260/280, and the extracted DNA was stored at −20 °C prior to PCR analysis.

## PCR assay

After DNA extraction, the quality of all DNA samples was confirmed by PCR using the host gene; receptor tyrosine-protein kinase (erbB-2). The reference sequences of erbB2 for three species (sheep, goat, and cattle) were obtained from GenBank. The sequences were aligned and the primers were designed according to the conserved region of all species. A set of forward erbB2-F (5′-AGAACCTGCGAGTAATCC-3′) and reverse primers erbB2-R (5′-CCTTCTCTTCACTATACATACAC-3′) were used to amplify a 128-base pair (sheep and goat) and a 126-base pair (cattle) of the erbB2 region. The DNA fragments of the mentioned region were amplified in a total volume of 25 μl containing 12.5 μl of Premix (Ampliqon, Odense, Denmark), 1 μl MgCl2 (25 mM), 0.75 μl of each primers (25 pmol), 3 μl template DNA and 7 μl Double-distilled water (DDW) in automatic thermo cycler (Bio-Rad C1000; Bio-Rad, Hercules, CA, USA) under the following conditions: 93 °C for 3 min as initial denaturation followed by 35 cycle at 93 °C for 30 s as denaturation, 58 °C for 30 s as annealing, 72 °C for 30 s as extension, and final extension at 72 °C for 3 min. Samples with 3 μl of DDW instead of DNA was used as Blank. 5 μl of the PCR products were analyzed through electrophoresis in a 1.5% agarose gel (BioNeer, Daejeon, South Korea) in 1X TBE buffer at 90 V for 30 min and were visualized and photographed on transilluminator (UVITEC).

### Toxoplasma

The detection of *T. gondii* was carried out using the amplification of a repetitive 529-bp DNA sequence (RE). This gene was selected as the target for PCR amplification and has been shown to be 200- to 300-fold more sensitive than other markers. The forward primer TOX-4 (5′-CGCTGCAGGGAGGAAGACGAAAGTTG-3′) and the reverse primer TOX-5 (5′- CGCTGCAGACACAGTGCATCTGGATT-3′) were used (*Homan et al., 2000*). PCR was amplified in a total volume of 25 μl containing 12.5 μl of commercial premix (Ampliqon, Odense, Denmark), 1 μl of total DNA, 0.6 μl of each primer (10 pmol/μl) (BioNeer, Daejeon, South Korea), and 10.3 μl of PCR H$_2$O. The reaction mixture was made in a thermocycler (Bio-Rad C1000, Bio-Rad, Hercules, CA, USA) with minor modification conditions: 93 °C for 5 min as the initial denaturation followed by 30 cycles at 93 °C for 30 s as denaturation, 55 °C for 30 s as annealing, 72 °C for 30 s as extension, and a final extension at 72 °C for 5 min. A negative control (1 μl deionized distilled water [DDW] instead of DNA) and a positive control (*T. gondii* DNA, accession number KT715444) were also included in each reaction.

### Neospora

For *N. caninum* molecular diagnosis, a fragment of the Nc5 gene was designed by nested PCR using Oligonucleotide primers Np21-plus (5′-CCCAGTGCGTCCAATCCTGTAAC-3′) and Np6-plus (5′-CTCGCCAGTCCAACCTACGTCTTCT-3′, as the external primer

pair) and Np6 (5′-CAGTCAACCTACGTCTTCT-3′) and Np7 (5′-GGGTGAACCGAGG-GAGTTG-3′, as the internal primer pair) (*Hughes et al., 2006*; *Müller et al., 1996*). All the PCR reactions were performed as was mentioned above for *T. gondii*. For the first round of PCR, the reaction mixtures were prepared in an automatic thermocycler (BioRad C1000, USA) under the following conditions: 94 °C for 5 min, followed by 40 cycles at 94 °C for 40 s, 62 °C for 30 s, 72 °C for 30 s, and a final extension at 72 °C for 10 min. The products of the first round were used as template for the second round of amplification, which was conducted under the following thermocycling conditions: 94 °C for 4 min, followed by 30 cycles at 94 °C for 30 s, 56 °C for 30 s, and 72 °C for 30 s. A final extension step was continued for another 3 min at 72 °C. Samples with 1 µl of DDW instead of DNA were used as negative controls and DNA of *N. caninum* (accession No.: kR106185) was considered as positive control.

### *Brucella*

To confirm *Brucella* species isolates, a previously described AMOS-PCR assay was applied (*Bricker & Halling, 1994*). Specific oligonucleotide primers were used targeting IS*711* (insertion sequence) in *Brucella melitensis* and *Brucella abortus* for molecular detection (*B. melitensis* primer: AAATCGCGTCCTTGCTGGTCTGA, *B. abortus* primer: GACGAACGGAATTTTTCCAATCCC and IS*711* primer: TGCCGATCACTTAAGGGC-CTTCAT) (*Bricker & Halling, 1994*). PCR amplification was carried out on 1 µl of genomic DNA (prepared freshly by the boiling method as described above) with the following steps: 95 °C for 5 min as initial denaturation followed by 35 cycles at 95 °C for 75 s as denaturation, 55.5 °C for 2 min as annealing, 72 °C for 2 min as extension, and final extension at 72 °C for 5 min. PCR mixture also contained 12.5 µl of 2× Master Mix (Ampliqon, Odense, Denmark), 0.5 µl of *B. abortus* and *B. melitensis* primers, 1 µl of IS711 primer (10 pmol/µl), and water up to a total volume of 25 µl. Also, 1 µl of DDW instead of DNA (Blank) and 1 µl of *B. melitensis* strain 16 M, and *B. abortus* strain 544 DNA (positive controls) were included in each reaction.

## Electrophoresis

In addition, 5 ml of the PCR products was run with electrophoresis through a 1% agarose gel (BioNeer, Daejeon, South Korea) stained with safe stain (0.5 µg/ml- Cina Gen Co, Iran) and visualized using a transilluminator (UVITEC).

## Sequencing of the PCR products

PCR products from 19 samples were purified using Gent Bio purification kit (Tabasmed, Tehran, Iran) and sequenced by Pishgam Company. The obtained sequences were edited and aligned using Sequencher Tmv.4.1.4 software.

## Statistical analysis

Statistical analysis was performed using Chi-squared test in SPSS, version 14.0. $P$-value less than 0.05 was considered statistically significant.

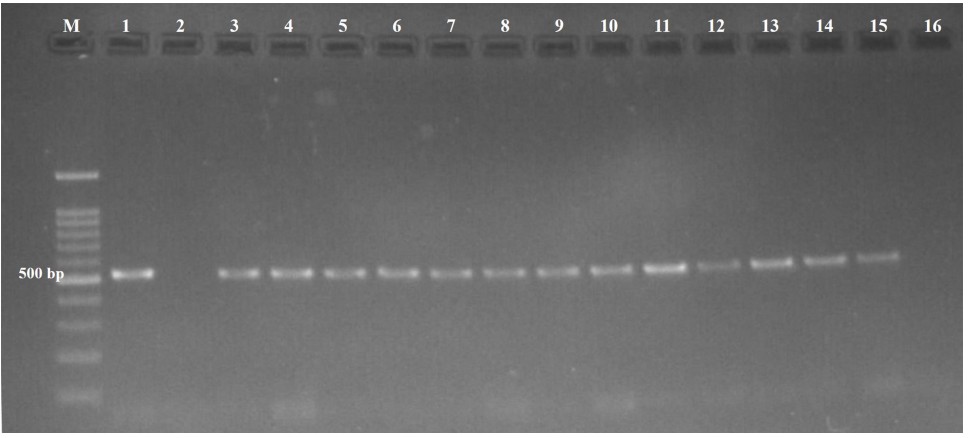

**Figure 1** **Agarose gel (1.5%) stained showing amplicons of *Toxoplasmagondii*.** Lane M, 100 bp DNA marker; Lane 1, Positive control; Lane 2, Negative control; Lane 3-15, Positive samples; Lane 16, Negative sample.

**Table 1** **Summary of culture and PCR results obtained in samples submitted to infected fetuses.**

| Species | No. samples | No. (%) of identified infections | | | | Total positive |
|---|---|---|---|---|---|---|
| | | *T. gondii* by PCR | *N. caninum* by PCR | *B. melitensis* by culture and PCR | *E. coli* by culture | |
| Sheep | 57 | 11 (19.3) | 2 (3.5) | 2 (3.5) | 2 (3.5) | 17 (29.8) |
| Goats | 4 | 1 (25) | – | – | – | 1 (25) |
| Cattle | 9 | 1 (11.1) | 2 (22.2) | – | 1 (11.1) | 4 (44.4) |
| Total | 70 | 13 (18.6) | 4 (5.7) | 2 (2.85) | 3 (4.3) | 22 (31.4) |

## RESULTS

Findings of our study are summarized in Table 1. Seventy samples (i.e., 57 sheep, four goats, and nine cattle) were obtained from aborted or dead fetuses from 14 counties in Mazandaran Province. The great majority of the abortions occurred during late gestation.

Protozoal infections were detected by specific PCR in 17 out of 70 (24.3%) examined fetuses. Of the infected fetuses, 22.8% (13/57), 25% (1/4), and 33.3% (3/9) were ovine, caprine, and bovine fetuses, respectively. The presence of *T. gondii* DNA (Fig. 1) was confirmed in 13 out of 70 fetuses (18.6%), and *N. caninum* DNA (Fig. 2) was detected in four out of 70 fetuses (5.7%) (Table 1).

Positive bacterial cultures were obtained from five fetuses, *Escherichia coli* (3 cases) and *Brucella* spp. (two cases). *Brucella* strains were identified as *Brucella melitensis* biovar 1 by the conventional methods. Both *Brucella* isolates produced a band of about 700 bp by PCR specific for *B. melitensis* (data for one isolate are shown in Fig. 3).

Nineteen samples from the aborted fetuses were subjected to sequencing followed by 20 µL of PCR product using the mentioned forward and reverse primers. The sequences obtained were verified by aligning them with the relevant sequences related

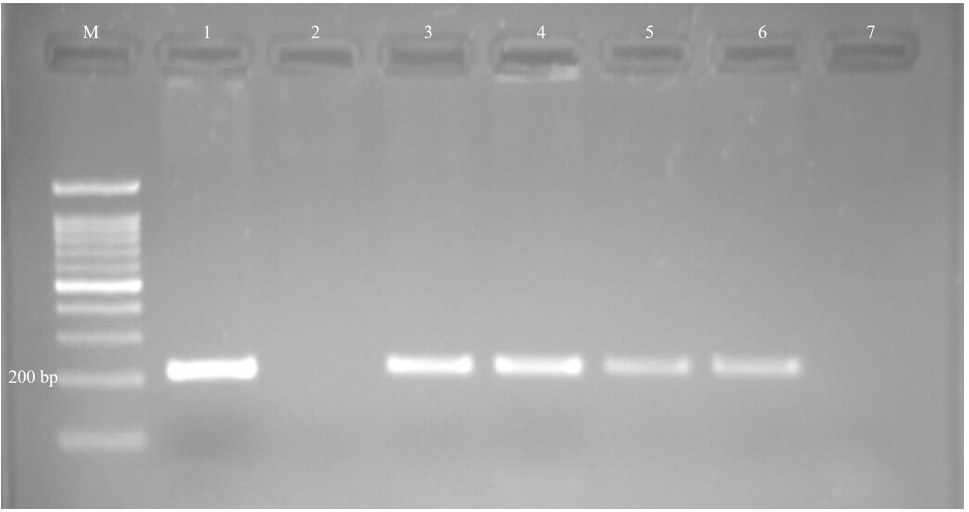

**Figure 2** **Examples of agarose gel electrophoresis of *Neospora caninum* obtained by nested-PCR.** Lane M, 100 bp DNA marker; Lane 1, Positive control; Lane 2, Negative control; Lane 3-6, Positive samples; Lane 7, Negative sample.

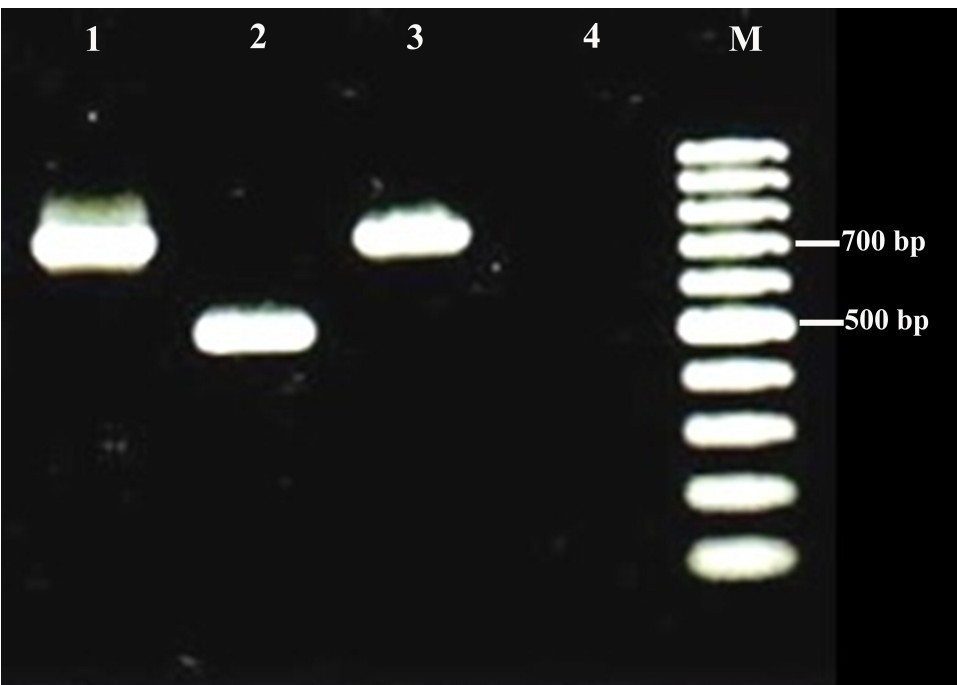

**Figure 3** **Examples of agarose gel electrophoresis of *Brucella* species PCR products using multiplex PCR.** Lane 1, *Brucella melitensis* strain 16 M (as positive control); Lane 2, *Brucella abortus* strain 544 (as positive control); Lane 3, *Brucella melitensis* biovar 1 isolate; Lane 4, Negative control (without template DNA); Lane M, 100 bp DNA marker.

**Table 2  Summary of identified infections status in three areas from Mazandaran providence, Northern Iran.**

| Area | No. samples | No. positive of identified infections (%) | | | | No. total positive (%) | OR | CI (95%) | P. value |
|------|-------------|------------|------------|-------------|---------|------------------------|-----|----------|----------|
| | | *T. gondii* | *N. caninum* | *B. melitensis* | *E. coli* | | | | |
| East | 16 | 1 (6.25) | 1 (6.25) | 0 | 0 | 2 (12.5) | 1 | – | – |
| Center | 39 | 12 (30.8) | 3 (7.7) | 0 | 3 (7.7) | 18 (46.2) | 0.16 | (0.01–0.9) | P = 0.02* |
| West | 15 | 0 | 0 | 2 (13.3) | 0 | 2 (13.3) | 0.9 | (0.05 - 14) | P = 0.9 |

**Notes.**
*Variable which displays significant difference ($p < 0.05$) using Chi-square test.

to *T. gondii* (GenBank accessions no. MH680820, MH884735 to MH884745, and MH884748), *N. caninum* (GenBank accessions no. MH841974, MH884746, MH884747, and MH884749), and *B. melitensis* (GenBank accessions no. MH687538 and MH687539).

## DISCUSSION

Abortion and fetal death can globally result in heavy economic losses to the livestock industry, which may be due to either infectious or non-infectious causes. The definitive diagnosis of abortifacient agents is often difficult in pasture-reared ruminants because few fetuses are usually available for laboratory evaluation. This finding could also be attributed to autolysis in these agents. The present study pinpoints that these pathogens may be causes of ruminant abortion in Mazandaran Province, north of Iran. Infectious agents were detected in 31.4% of the examined cases. The rate of positive results was higher in the samples from the central area relative to those in other areas, and it seems that contamination to at least one pathogen might be higher in this area than others (Table 2). The aetiologic agents associated with abortion were determined in previous studies using various diagnostic techniques (*Campero et al., 2003*; *Kim et al., 2002*; *Moreno et al., 2012*; *Pereira-Bueno et al., 2004*).

For the diagnosis of protozoan causes of abortion in livestock, a wide range of techniques have been applied including serological and histopathological methods, immunohistochemistry, bioassay, cell culture, and molecular assays. PCR method is considered a specific and sensitive technique for the detection of parasite-specific DNA sequences (*Wastling, Nicoll & Buxton, 1993*). We used a 200- to 300-fold repetitive 529-bp fragment for the diagnosis of toxoplasmosis because of its high sensitivity and specificity (*Homan et al., 2000*). Also, for the evaluation of neosporosis, we selected the nested PCR technique based on the highly repeated Nc5 region with the 5-fold sensitivity of detection (*Almería et al., 2002*). The current study indicated that protozoan infections are major causes of abortion in animals. According to molecular examination of the brain samples taken from fetal samples, *T. gondii* and *N. caninum* were detected in 13 (18.6%) and 4 (5.7%) cases, respectively. The highest prevalence of *T. gondii* and *N. caninum* infections were in ovine and bovine fetuses, respectively (Table 1). These findings are in agreement with those of other studies where these parasites were implicated in the majority of abortions in the mentioned animal hosts (*Kim et al., 2002*; *Masala et al., 2007*). Our work showed lower prevalence of *N. caninum* than *T. gondii,* which may be attributed to the fact

that *N. caninum* is considered as one of the most important causes of reproductive failure in cattle (*Dubey, Schares & Ortega-Mora, 2007*) whereas in the present study, the majority of the tested animals were sheep. Protozoan infections associated with abortion are often reported in the literature. *Habibi et al. (2012)* showed higher molecular prevalence rates of *T. gondii* (ovine abortions: 37.5% and caprine abortions: 22.7%) than this study.

*Moreno et al. (2012)* examined 74 ovine and 26 caprine fetuses for the presence of *N. caninum* and *T. gondii* DNA in Spain, respectively, and they showed that their prevalence rates were 5.4% and 6.8% in ovine abortions and 3.8% and 11.5% in caprine abortions, respectively. In Switzerland, Sagar et al. reported *T. gondii* DNA in 1 (<1%) of the 242 aborted bovine fetuses, and *N. caninum* DNA was detected in 50 (21%) of them (*Sager et al., 2001*). These differences may be attributed to the geographical distribution and employed techniques in the diagnosis of infection. Moreover, our findings indicated that the samples were not co-infected with both *T. gondii* and *N. caninum*, but further studies with larger sample sizes are required on this issue.

In addition to protozoan agents, *B. melitensis* and *E. coli* were also isolated from fetal samples as bacterial infectious causes of abortion. Two ovine samples were detected with *B. melitensis* infection. This was supported by both conventional bacteriological and molecular methods. Although positive culture is considered the gold standard test for the definitive diagnosis of brucellosis (*Araj, 2010*), PCR assay was applied for confirmation of the diagnosis at the species level (*Bricker & Halling, 1994*). *B. melitensis* biovar 1 was found in 2 (2.9%) out of 70 examined samples taken from infected herds in Babolsar, Iran, with high sero-prevalence according to serological surveys using RBT, SAT, and 2ME. However, the interpretation of serological responses to *Brucella* is difficult because of false positive results due to the mass vaccination campaign in the region, therefore, the findings should be verified by bacterial isolation (*Fekete, Bantle & Halling, 1992*). It should also be mentioned that *B. melitensis* biovar 1 was the most prevalent *B. melitensis* in Mazandaran Province (*Behroozikhah et al., 2012*). In other studies, *Brucella* was detected in 20.86%, 31%, and 34.56% of aborted sheep fetuses in Iran, Turkey, and Greece, respectively. Our finding is in accordance with the results of studies that reported *B. melitensis* as a prevalent cause of abortion in sheep (*Dehkordi, Saberian & Momtaz, 2012*; *Leyla, Kadri & Ümran, 2003*; *Samadi et al., 2010*). Immunization with *Brucella melitensis* strain Rev.1 is known as the most practical strategy for brucellosis management in the low socioeconomic conditions prevailing over Mazandaran Province, which is currently performed by Iranian Veterinary Organization. However, culling infected animals following serological testing, as well as selecting and replacing animals from brucellosis-free flocks can accelerate disease elimination (*Blasco & Molina-Flores, 2011*; *World Organization for Animal Health , OIE*). While the vaccination campaign is limited to young animals at present, implementing test-and-slaughter one year post-vaccination might slightly decrease false positive serologic results induced by the vaccine which make this strategy feasible (*World Organization for Animal Health , OIE*).

Moreover, *E. coli* from three ovine fetal samples was isolated in pure culture, when abomasal contents were cultured on blood, MacConkey, and EMB agars and incubated aerobically for 24–48 h. Organisms produced pink colonies on the MacConkey agar and

green metallic sheen colonies on the EMB agar plates (*Borel et al., 2014*; *Quinn et al., 2011*). *Salmonella* spp. was not detected in our study, which suggests that future surveys are required to elucidate their contribution to fetal mortality in the province. In the present study, no fungi were identified as etiological agents, which could be due to the sporadic nature of fungal abortion.

## CONCLUSIONS

Our results suggest that infection with the studied pathogenic agents occurs in ruminant herds/flocks in Mazandaran, Iran, resulting in fetal mortality. Thus, by promoting sanitary animal production practices, health education programs can reduce the transmission of infectious agents to humans and farm animals. Moreover, an appropriate immunization strategy using a proper *Brucella* vaccine could diminish reproductive losses in livestock. Further studies are necessary to investigate the precise epidemiology and prevalence rates of these causative agents of abortion in ruminants and evaluate the resulting economic losses to the industry in the province.

## ACKNOWLEDGEMENTS

We express our thanks of the Central Laboratory of the Department of Veterinary Medicine in the Mazandaran province for providing samples. The authors thank Deputy of Research of Mazandaran University of Medical Sciences for their excellent supervision of this Project (No. 1055). We wish to thank Saeid Salehi and Mohammad Naghi Rahimi for their kind help during this research.

### Funding
The authors received no funding for this work.

### Competing Interests
The authors declare there are no competing interests.

### Author Contributions
- Afsaneh Amouei and Azadeh Mizani conceived and designed the experiments, performed the experiments, contributed reagents/materials/analysis tools, prepared figures and/or tables, authored or reviewed drafts of the paper, approved the final draft.
- Mehdi Sharif, Shahabeddin Sarvi, Sargis A. Aghayan, Mohammad Bagher Hashemi-Soteh and Ahmad Daryani conceived and designed the experiments, contributed reagents/materials/analysis tools, authored or reviewed drafts of the paper, approved the final draft.
- Ramin Bagheri Nejad, Sara Gholami, Alireza Sadeghi and Mohammad Sarafrazi conceived and designed the experiments, performed the experiments, contributed reagents/materials/analysis tools, authored or reviewed drafts of the paper, approved the final draft.

- Seyed Abdollah Hosseini conceived and designed the experiments, analyzed the data, contributed reagents/materials/analysis tools, prepared figures and/or tables, authored or reviewed drafts of the paper, approved the final draft.

## Animal Ethics

The following information was supplied relating to ethical approvals (i.e., approving body and any reference numbers):

All investigations reported here were approved by the Ethics Committee of Mazandaran University of Medical Sciences (No. 1055).

## Data Availability

The raw data are provided in a Supplemental File.

## Supplemental Information

Supplemental information for this article can be found online at http://dx.doi.org/10.7717/peerj.5920#supplemental-information.

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
