# Peer review of "Aetiology of livestock fetal mortality in Mazandaran province, Iran"

_PeerJ, doi:10.7717/peerj.5920_

## Round 0.1 · original submission · Major Revisions

Please note that a major revision is requested, which means that you are expected to take into account ALL of the reviewers comments. The reviewers make a number of good points that you will need to address in your resubmission. In addition all 3 reviewers picked up on the English not being of sufficient standard in the current version. As the Journal does not provide an editing service, it is important that you improve this aspect of your submission. So please find an a appropriate person to help with this. Failure to do this will result in the manuscript being rejected.

Reviewer 1 ·

Basic reporting

1) The English language usage is poor. It needs to be carefully revised throughout the entire manuscript. It is often difficult to understand the authors’ point, particularly in the discussion section.
2) The figures are not relevant to the content of the article. The 3 figures only show the electrophoresis of positive and negative PCR controls and one example of a positive and a negative sample.
3) Table 2 presents, per region, the total of samples analyzed and the number of samples that were positive to one of the pathogens tested, not separated per pathogenic agent. The authors use this table to base the claim that “The fetuses of central area showed higher positive level than the other areas and it seems that this region might be more contaminated” (lines 2014-105) but this information is not relevant when 4 completely different pathogenic agents are being taken together.

Experimental design

no comment

Validity of the findings

no comment

Additional comments

The manuscript reports on the detection of T. gondii, N. caninum, B. melitensis and E. coli in aborted fetus from sheep, cattle and goat from Mazandaran province, Iran. The purpose of the study is of great interest. However, the manuscript contains too limited information to justify publication.
A few further comments:
1) The number of samples, 70 aborted fetus from sheep (57), cattle (9) and goat (4), considering the description of the work that is presented, is too small to draw other conclusions than that these pathogenic agents occur in the area studied. However, it is possible that a detailed analysis of the results obtained taking into consideration the origin of the samples at farm level, may show interesting results. For example, just to make my point clearer, the 10 T. gondii positive results in sheep from the Center region have different meanings if all came from one farm or from different farms.
2) Samples of brain tissue and abomasal content were obtained from aborted, stillborn or weak animals that died (lines 108-109). However, only results from fetus are presented. It may be interesting to present and analyze the other results even if they are all negative.
3) There are numbers between brackets in the manuscript that probably correspond to citations of bibliographic references in a format that doesn’t have correspondence in the References section.
4) The claim that the gene chosen for T gondii PCR detection has been shown to be 200- to 300-fold more sensitive than other markers (lines 138-139), if it is not a mistake, needs bibliographic support.
5) The “C” in degree Celsius symbol should not superscript.

Reviewer 2 ·

Basic reporting

The authors describe a current topic of importance to the livestock sector and with implications in public health.

Generally, the literature references are chosen well, but do not always comply with the journal's guidelines (numbers are used in the text body while the reference list is in alphabetical order without numbering).

Raw data and figures are appropriate.

The written English should be improved throughout the manuscript. Especially in the discussion section the English is weak and partly incomprehensible.

Experimental design

The methods the authors used are sound and properly described; however, it has to be noted that for Brucella spp. differentiation improvements of the chosen PCR method (AMOS-PCR) have been described (Bricker and Halling, 1995; Ewalt and Bricker, 2002).

The interpretation of the findings from bacterial culture are not very clear:

- Which criteria were followed to determine E. coli as etiological agent of the abortion? For example, was it found in pure culture, in fetal organs, were other infectious agents excluded, check e.g. Borel et al. 2014.

- Which specific culture was done for Salmonella spp. detection? Generally, Salmonella are detected better using enrichment media, especially in heavily contaminated samples as is abortion material. If this was not the case, I would suggest to remove the discussion about Salmonella not being an important etiological agent.

Validity of the findings

The impact and novelty of this study is geographically limited to Mazandaran province.
While the aim of the study is clear, the methods valid and the results comprehensible, the conclusion should be more concise. Which immunization would the authors recommend? What are the concrete measures for improved biosecurity and why is it suggested as an alternative to immunization? Would the authors apply these measures restricted to the restricted study area?

Additional comments

Specific comments:
- Throughout the manuscript please make sure of the correct abbreviation of degree Celsius °C with capital C.

- line 83: "some of the species of the genus Brucella are pathogenic for human (Leyla et al., 2003)" Although there are species and biovars that have been discussed as less pathogenic or even apathogenic to humans, I would rather state that the genus Brucella is generally considered pathogenic for humans (=zoonotic). For further reading see e.g. WHATMORE A.M. (2009). Current understanding of the genetic diversity of Brucella, an expanding genus of zoonotic pathogens. The chosen reference seems somewhat inappropriate.

- line 99: which tests are you exactly referring to with "Wright" (SAT?) and "2ME", please specify and use references. The current version of the OIE Manual Chapter 2.1.4. from May 2016 provides a good overview over the current methods. Moreover, first time you use 2ME please spell out (2-mercaptoethanol).

- line 114: comply to reference style

- line 126: genus names "Toxoplasma" and "Neospora" in italics

- line 129: "DNA....prepared by boiling method." I would suggest to replace this by "PCRs were performed on heat-killed cell suspensions. For this purpose, a loopful of bacterial cells..."

- line 144: H2O, please use subscript.

- line 148: please replace "acted" with "included"

- lines 164-177: the PCR you describe is commonly known as AMOS-PCR. Please italicize the numbers (711) following IS (Insertion Sequence). Please be aware that this PCR has been continously improved. While your method is valid, you could include advantages of using a more updated method in the discussion, e.g. Bricker and Halling 1995) that is able to differentiate between field and two vaccine strains.

- line 187: fourteen = 14

- line 188: with "last months of gestation" do you mean the last trimester?

- lines 198-252: The discussion is written in weak English and sometimes not comprehensible at all. It needs substantial improvement.

- line 206: please replace "researches" by "studies"

- lines 250-252: regarding E. coli and Salmonella please refer to my comments in the section "2. experimental design". Regarding fungal abortion, which is known to occur only sporadically in up to 2% of abortions, it is considering the number of cases analyzed (=70) not safe to state that fungi are not important. It would be sufficient to state that no fungi were identified as etiological agents which could be due to the sporadic nature of fungal abortion.

- lines 254-263: the conclusion should be more concise. The advantages and disadvantages of vaccination should be discussed (e.g. interference with diagnostics). Which vaccines are available? Why immunization "or" biosecurity? This should be included in the general discussion. Please also refer to my comments in the section "3. Validity of the findings".

Reviewer 3 ·

Basic reporting

The manuscript requires some attention to the english throughout to help with the understanding.

There is sufficient context and relevant literature provided for the paper.

I am not so familiar with the layout of the journal but would suggest that the data could be written up as a short communication?

Experimental design

The research is well defined and the methods used are described with sufficient detail and information to replicate.

Validity of the findings

This study reports the investigation of 70 aborted foetuses collected from sheep (n=57), goats (n=4) and cattle (n=9) looking for the presence of T.gondii, N.caninum and Brucella spp as these are known to be common causes of infectious abortion in ruminant species.
The results show the presence of T. gondii DNA in around 20% of the aborted ovine foetuses suggesting that T. gondii may be an important cause of abortion in sheep. In future studies the authors might wish to do some histopathology analysis of the tissues to look for lesions associated with T. gondii infections to show that T. gondii is acting as a primary pathogen in this context.
There were fewer samples available for goats and cattle, although interestingly N. caninum DNA was detected in the bovine foetal samples and this parasite is known to be an important cause of abortion in cattle. Vertical transmission of N.caninum is very common and does not always result in abortion, therefore the presence of N.caninum DNA shows that the parasite is present but does not necessarily mean it was the cause of abortion. Some further histopathology testing to look for specific lesions will give further information in this regard.
The authors might also wish to consider examining foetal serology to give additional diagnostic information.

Additional comments

The English language needs some attention

---

## Round 0.2 · Minor Revisions

Please address the remaining concerns that the reviewer raises relating to the interpretation of your findings and making sure to only make statements which are supported by the results.

Reviewer 1 ·

Basic reporting

no comment

Experimental design

no comment

Validity of the findings

no comment

Additional comments

The manuscript was significantly improved and is now clear. I have no major comments on the present version, apart from the limited amount of novelty it reports.
The work is based on 70 aborted fetuses from Mazandaran province and their origin is presented as from east (n=16), center (n=39), and west (n=15) of the province. T. gondii was found in 1 aborted fetuses from east and 12 from center, N. caninum in 1 from east and 3 from center, B. melitensis in 2 from west and E. coli in 3 from center.
Considering the information provided, only the occurrence of these pathogenic agents can be concluded.
Sentences as, for example, “Our results showed that infection with the mentioned pathogenic agents brings about considerable fetal mortality” (abstract, lines 46-48) or “The present study pinpoints the main causes of ruminant abortion in Mazandaran Province” (discussion, line 212) do not find support in the manuscript.

---

## Round 0.3 · Major Revisions

Dear Dr Amouei,

Thank you for your appeal letter of the 27th May 2018.

After having reviewed your manuscript again at your request and taking advice I am pleased to let you know that we consider a revised manuscript for publication if the following 2 criteria (that were criticisms of the review process) are met:

1. You demonstrate that you can amplify by PCR a housekeeping gene from all DNA samples used in the study.

2. You sequence each positive PCR product to demonstrate that it is the expected product.

Yours sincerely,

Craig

· Appeal

Appeal

Dear Editor in Chief

As you know we have been already submitted our paper (20386) entiteled “Aetiology of livestock fetal mortality in Mazandaran province Iran” since September 2017. Then based on you and your reviewers, it has been under the second revision and we did all corrections which you asked.

However, unfortunately after 9 months you decided as rejection.

We believed you did very well job and we respect to your decision.

But I was wondering regarding those questions which you asked, would you please take into considerations.

Firstly, there is concern that negative results are caused by degraded DNA. A control for DNA quality is missing, this would be essential. This is necessary as material cannot always be analysed right away in such studies.

Reply: Thanks for critical question and we are agreeing, however as you aware we used DNA control which obtained from RH strains and all DNA samples have been checked with both Nano drop and gel electrophoresis to be sure for quality and quantity.

Secondly, it was felt that PCR analyses alone are not enough for the analysis presented. Without sequencing data of the PCR products we cannot rule out that some aberrant genes have been amplified, providing false positive results.

Reply: I am completely agreed with you. We did PCR analysis, however we have this opportunity to do DNA sequencing for confirmation and then we will deposit to GenBank data base. I would be interesting to have their accession number for better understanding.

Thirdly, the microbiology data could be strengthened using microbiome analysis (NGS).

Reply: Again thanks for critical question and honestly I would confess that it would be nice to have NGS data, but in this point we cannot do this experiment, because it is very expensive to do. However, it is up to you.

Again thanks for consideration and we are looking forward to hearing from you.

With Best regards

Afsaneh Amouei


· · Academic Editor

Reject

After consultation withe a Section Editor for this area of PeerJ, it was decided that the study as it stands had some issues that preclude publication in PeerJ.

Firstly, there is concern that negative results are caused by degraded DNA. A control for DNA quality is missing, this would be essential. This is necessary as material cannot always be analysed right away in such studies.

Secondly, it was felt that PCR analyses alone are not enough for the analysis presented. Without sequencing data of the PCR products we cannot rule out that some aberrant genes have been amplified, providing false positive results.

Thirdly, the microbiology data could be strengthened using microbiome analysis (NGS).

---

## Round 0.4 · accepted · Accept

Thank you for you very much for performing the extra work. I have annotated a few changes to the text on the attached PDF which you can address while in production.

#